# Characteristics and Antioxidant Activity of Walnut Oil Using Various Pretreatment and Processing Technologies

**DOI:** 10.3390/foods11121698

**Published:** 2022-06-09

**Authors:** Pan Gao, Yunpeng Ding, Zhe Chen, Zhangtao Zhou, Wu Zhong, Chuanrong Hu, Dongping He, Xingguo Wang

**Affiliations:** 1Key Laboratory for Deep Processing of Major Grain and Oil (Wuhan Polytechnic University) of Ministry of Education in China, College of Food Science and Engineering, Wuhan Polytechnic University, 68 Xuefu South Road, Changqing Garden, Wuhan 430023, China; dyp99112@163.com (Y.D.); 18162571387@163.com (Z.Z.); zhongwu@whpu.edu.cn (W.Z.); hcr305@163.com (C.H.); hedp123456@163.com (D.H.); wangxg1002@gmail.com (X.W.); 2Key Laboratory of Edible Oil Quality and Safety for State Market Regulation, Wuhan Institute for Food and Cosmetic Control, 1137 Jinshan Avenue, Wuhan 430012, China; whpuchenzhe@163.com; 3National Engineering Research Center for Functional Food, School of Food Science and Technology, Jiangnan University, 1800 Lihu Road, Wuxi 214122, China

**Keywords:** walnut oil, pretreatment, processing, subcritical butane extraction, steaming roasting

## Abstract

This study was the first time the effects of pretreatment technology (microwave roasting, MR; oven roasting, OR; steaming roasting, SR) and processing technology (screw pressing, SP; aqueous enzymatic extraction, AEE; subcritical butane extraction, SBE) on the quality (physicochemical properties, phytochemical content, and antioxidant ability) of walnut oil were systematically compared. The results showed that the roasting pretreatment would reduce the lipid yield of walnut oil and SBE (59.53–61.19%) was the processing method with the highest yield. SR-AEE oil provided higher acid value (2.49 mg/g) and peroxide value (4.16 mmol/kg), while MR-SP oil had the highest content of polyunsaturated fatty acid (73.69%), total tocopherol (419.85 mg/kg) and total phenolic compounds (TPC, 13.12 mg/kg). The DPPH-polar and ABTS free radicals’ scavenging abilities were accorded with SBE > AEE > SP. SBE is the recommended process for improving the extraction yield and antioxidant ability of walnut oil. Hierarchical cluster analysis showed that processing technology had a greater impact on walnut oil than pretreatment technology. In addition, multiple linear regression revealed C18:0, δ-tocopherol and TPC had positive effects on the antioxidant ability of walnut oil, while C18:1n-9, C18:3n-3 and γ-tocopherol were negatively correlated with antioxidant activity. Thus, this a promising implication for walnut oil production.

## 1. Introduction

Walnuts (*Juglans regia* L.) are widely cultivated and important oilseed crops in China [1]. Chinese walnut production is 1.78 million tons and its output is 4795.9 kilotons, which accounts for half of the world’s walnut production (2020, FAO stat). Walnuts have been eaten in China for thousands of years, influenced by the traditional concept of “shape compensation”, as the kinds of nuts that have benefits for brain health [2]. This speculation has also been demonstrated in walnut oil; a recent date showed that walnut oil could reduce memory impairment in mice [3] because walnut oil has anti-inflammatory properties [4]. In recent years, the research on walnut oil has become popular because walnut oil is a kind of edible oil with high nutritional value. Walnut oil nutrition research mainly focuses on the role of walnut oil in intestinal diseases, which is widely used in traditional medicine around the world and is prescribed as beneficial food oil in agroindustry [5]. Walnut oil is a prominent functional food candidate for inflammatory bowel disease treatment [6] and ulcerative colitis treatment [7] because it has good antiaging activity in vivo [8] and can increase antioxidant capacity [9]. The anti-inflammatory action is more dominant in the cases of ameliorating inflammatory bowel disease and ulcerative colitis. Moreover, for the above-mentioned health outcome of preventing memory impairment, antiaging and antioxidative potentials would serve more beneficial roles. The antioxidant capacity of walnut oil is closely related to its processing technology [10]. Therefore, the reports related to walnut oil processing have attracted extensive attention.

Walnut oil processing traditionally focuses on organic solvent extraction [11,12], and it has high volatility, toxicity and flammability [13], which usually involve immense environmental pollution, human health risks and high operating costs [14]. Thus, growing attention has been attracted to developing an environmentally friendly method of extracting walnut oil in recent years. Screw pressing (SP) can produce high-quality oils, be environmentally friendly and require less energy than organic solvent extraction [15]. However, its main disadvantage is that it generates low extraction yields, which limits its industrial use [16]. Aqueous enzymatic extraction (AEE) is a promising methodology since it is ecofriendly and provides healthful nutrition [17]. In addition, the oil obtained is of better quality because it does not present organic solvents or antinutritional compounds (3,4-benzypyrene and other polycyclic aromatic compounds) and the walnut meal is also of superior quality for human and animal consumption [13]. The subcritical butane extraction (SBE) method is another environmentally friendly process method, which is also proven to reduce the degradation of the bioactive components, resulting in a final extracted product that is free of toxic residual solvents [18]. However, AEE and SBE have several disadvantages, such as high costs and low lipid yield [19]. Our previous study [20] proved that roasting pretreatment could improve the lipid yield and bioactive components of walnut oil. Therefore, to improve oil extraction yields or nutrition value, suitable roasting pretreatment methods should be used.

The research on the roasting pretreatment methods of walnut oil focuses on roasting conditions [21,22,23]. In addition, the effects of walnut oil produced by microwaves and soaking on antioxidant and antiproliferative activity have been compared [24]. Microwave roasting (MR) pretreatment has been proven to improve the oil extraction yield of yellow horn seed kernels [25] and camellia oleifera seed [26]. There are significant differences between MR and oven roasting (OR): the volatile compounds of camellia seed oil prepared by MR and OR are different [27] and the bioactive composition of orange seed oil pretreated by these two methods have differences [28]. In addition, steaming roasting (SR) is also the conventional processing pretreatment of flaxseed oil [29] and camellia seed oil [30]. The advantages of this pretreatment are preventing oil oxidation with an oxygen-free environment and enhancing thermal degradation by increasing heat transfer [31]. Notably, the study of steaming technology to pretreat walnuts and its effect on the nutritional components of walnut oil are scarce.

There are many small walnut oil production enterprises in China and pretreatment and processing methods are various. Therefore, it is very important to systematically compare the effects of roasting pretreatment and processing methods on the quality of walnut oil. In the present study, walnut oil produced using different pretreatment and processing methods was analyzed, including the lipid yield, physicochemical properties, fatty acid composition, phytochemical content, and antioxidant capacity. To guide the production of walnut oil with high nutritional value and better antioxidant capacity, the effects of pretreatment and processing methods on the quality of walnut oil were compared by hierarchical cluster analysis (HCA). Moreover, the main functional substances responsible for the antioxidant capacity of walnut oil were screened by multiple linear regression (MLR). It can be clarified that the impact of pretreatment and processing technologies on the chemical composition of walnut oil obtained the best processing technology of walnut oil and promoted the development of the related industry.

## 2. Materials and Methods

### 2.1. Materials

The walnuts were collected from the walnut planting base of Hubei Guicuiyuan Technology Co., Ltd. (Xiangyang, China). The walnut variety is Qingxiang, which was harvested during the 2020–2021 season. The mean value of annual precipitation is about 24.0 to 34.0 mm, air temperature ranges between 10.2 and 12.3 °C, the number of frost-free days is 200 to 230 and the sunshine duration is 2683.5 to 3167.1 h. The walnuts after harvest were transported to the laboratory, dried in an oven at 40 °C for 72 h to make their moisture content less than 8% and processed to walnut oil immediately. Walnut shells were broken by hand, and the kernels were separated from the shells before processing.

### 2.2. Chemicals and Standards

Standards of 37-fatty acid methyl esters, 2,2-Diphenyl-1-picrylhydrazyl (DPPH), 2,4,6-Tris (2-pyridyl)-*S*-triazine (TPTZ), 2,2′-Azino-bis (3-ethylbenzothiazoline-6-sulfonic acid) diammonium salt (ABTS) and 6-hydroxy-2,5,7,8-tetramethylchroman-2-carboxylic acid (Trolox) were purchased from Sigma-Aldrich Chemical Co. Ltd. (Shanghai, China). Standards of tocopherols (α-, β-, γ- and δ-tocopherol, purity > 95%), 5α-cholestane, campesterol, stigmasterol and β-sitosterol were provided by Aladdin Chemical Co. Ltd. (Shanghai, China). Cellulase (EC 3.2.1.1, ≥ 400 U/mg) and pectinase (EC 3.2.1.15, ≥ 500 U/mg) were purchased from Pangbo Biological Engineering Co., Ltd. (Nanning, China). Other solvents were obtained from Macklin Reagent Co., Ltd. (Wuhan, Hubei, China).

### 2.3. Oil Extraction

#### 2.3.1. Roasting Pretreatment

MR: Walnut kernels (500 g) were treated by a microwave-assisted extraction apparatus (Media, Guangzhou, China) for 10 min by applying a 1 min-pause mode, 1 min-run mode action with the condition of microwave power 540 W at a frequency of 2450 MHz.

OR: Walnut kernels (500 g) were subjected to roasting in an oven (GZX-9080, Boxun, China) under 160 °C for 10 min.

SR: Walnut kernels (500 g) were put into a steam generator (FY50, Sanshen, China) under 160 °C for 10 min at 0.54 MPa.

#### 2.3.2. Processing

SP: The walnut samples were pressed at room temperature using a ZJ-707 screw press (Wenfeng, Dongguan, China) to afford the oil, and 10% walnut shell was used as filler. The oil was centrifuged at 4000× *g* for 20 min at 4 °C.

AEE: The method was according to Cheng et al. [32] with some modifications. The samples were ground into a fine powder and commixed with distilled water at a ratio of 1:5 (*w/v*); the slurry was heated to 50 °C, held for 30 min and then cooled to the enzymatic digestion temperature. The cellulase and pectinase ratio 1:1 (*w/w*) was added and the mixture was agitated for 30 min at 40 °C. Enzymatic hydrolysis proceeded for a further 10 min at 90 °C under constant horizontal shaking in a Maxi Mix III rotary shaker (Thomas Scientific, Shanghai, China). The suspension was then centrifuged at 4000× *g* for 20 min at 4 °C.

SBE: The method was according to our previously published paper [10]. The samples were ground into a fine powder and extracted by subcritical extraction equipment (Zhengzhou, Henan, China) at 45 °C for 60 min in 0.5 MPa with a butane-to-kernel ratio of 7:1 (*v/w*). The oil was centrifuged at 4000× *g* for 20 min at 4 °C.

### 2.4. Physicochemical Properties

The acid value (AV) and peroxide value (PV) were determined according to the Cd 3d-63 and Cd 8b-90 official recommended by AOCS. The oil yield was recorded by the mass of oil extracted and the weight of the walnut kernel at room temperature (25 °C). The continuous bubbling of air at a flow rate of 20 L/h through oil samples (3.0 g) held at 120 °C was used to measure the oxidation stability index (OSI) with a Rancimat 892 model (Metrohm, Switzerland), which was expressed in hours.

### 2.5. Fatty Acid Composition

The fatty acid composition was carried out by the Agilent 7890A gas chromatography (GC) system (Agilent, CA, USA) by Gao et al. [20] with some modifications. The oil of 0.20 mg was methylated under alkaline conditions and analyzed by a DB-5 capillary column (30 m × 0.25 mm, 0.25 μm, Agilent, CA, USA). The GC adopted a programmed temperature rising mode, with an initial temperature of 60 °C for 3 min and then a temperature of 170 °C at a rate of 5 °C/min for 5 min and a temperature of 220 °C at a rate of 2 °C/min for 10 min. The carrier gas was helium (99.999%) with a flow rate of 1.0 mL/min, the inlet temperature was 250 °C, the injection volume was 1 μL, and the split ratio was 100:1. The fatty acid methyl ester peaks were identified by comparison to the retention times of known standards, and the percentage of each peak area to the sum of all peak areas was quantified.

### 2.6. Phytochemicals Content

The determination of phytosterols, tocopherols and squalene in pretreatment was as described by Liu et al. [33]. The mobile phase for HPLC-DAD (Ultimate 3000, Thermo Fisher, Waltham, MA, USA) analysis used methanol. The samples were injected onto a C18 Column (5 μm, 150 mm × 2.1 mm; ZORBAX Eclipse Plus, Agilent, CA, USA). The flow rate was 0.7 mL/min with a 10 μL injection volume and maintained the column at 30 °C. The quantification of the analytes by DAD and the wavelengths were 210 nm. The phytochemicals were identified by comparison with authentic standard retention times, and the results were expressed in mg/kg.

A solid phase extraction column (Sepax Technologies, Inc., Newark, DE, USA) and a Folin–Ciocalteu reagent method were used to analyze the content of total phenolic compounds (TPC), which was according to our previous report [9]. According to the gallic acid standard, the TPC content in each walnut oil sample was calculated, and the results were expressed in mg/kg.

### 2.7. Free Radical Scavenging Capacity

The 2.5 g of walnut oil was added to 5 mL of methanol and shaken 3 min in the dark. The supernatant was separated as the polar extract, and the remainder was retained as the nonpolar extract. Approximately 0.12 g of the walnut oil sample was used as a whole oil for the DPPH assay. Details of the employed methods can be found in our previous paper [10].

### 2.8. Statistical Analysis

All samples were analyzed in triplicate, and the results were expressed as means ± standard deviations (SDs). Statistical analysis was carried out using SPSS 23.0 (IBM, Armonk, NY, USA). For all evaluated parameters (Duncan’s multiple-range tests), ANOVA test results were considered to be statistically different at a level of 5%. Prior to regression analysis, to remove the influence of independent variable units, all independent variables were standardized (converted to z-scores). Hierarchical cluster analysis (HCA) was assessed based on the squared Euclidean distance, and centroid clustering was used to group the samples. Multiple linear regression (MLR) was conducted using a stepwise method.

## 3. Results and Discussion

### 3.1. Lipid Yield

Table 1 lists the lipid yields of walnut oils. The results showed that the lipid yield of walnut oil processing decreased in the order of SBE (59.53–61.19%) > AEE (50.64–53.65%) > SP (40.73–46.31%). This may be because SBE has the lowest loss of lipids in a sealed environment. The highest oil content was that of OR-SBE oil, which was 14.88% higher than that of the control (46.31%). However, the effect of pretreatment on the yield had no obvious regularity. The pretreatment decreased the lipid yield of SP oil; the average yield of SP oil samples with pretreatment was 41.60%, which was 4.71% less than that of the control. It proved that pretreatment had a negative effect on the processing of walnut oil by SP. The lipid yield of AEE oil in our study was lower than that of the report (75.4%) [34], which might be because the reaction conditions of AEE were not optimized. The lipid yield of AEE oil would improve by properly optimizing the experimental conditions.

### 3.2. Physicochemical Properties

The physicochemical properties of walnut oils are shown in Table 1. The AV of oils ranged from 0.14 to 2.49 mg KOH/g, which was lower than the standard of walnut oil in China (≤3 mg KOH/g). The AV of AEE walnut oil was the highest, with an average value of 1.58 mg KOH/g, which was much higher than that of SP (0.35 mg KOH/g) and SBE (0.24 mg KOH/g). Similar to AV results, the PV of SR-AEE oil was also higher (4.16 mmol/kg) than that of other samples. This might be because the triglycerides were broken down into free fatty acids under the action of enzymes, which would produce more free fatty acids, resulting in higher AV and PV [35]. The PV of the control (5.56 mmol/kg) was higher than that of others, which proved that the pretreatment processing could significantly reduce the PV of oils.

### 3.3. Fatty Acid Composition

Table 1 also shows the fatty acid composition and content of walnut oils. The result was obtained that the pretreatment and processing methods could not change the composition of fatty acid. The main fatty acids of walnut oil were palmitic acid (C16:0, 6.14–6.41%), stearic acid (C18:0, 2.62–2.77%), oleic acid (C18:1n-9, 17.36–18.84%), linoleic acid (C18:2n-6, 61.73–64.95%) and linolenic acid (C18:3n-3, 8.75–10.33%). The C18:1n-9 of AEE oils (average value 18.57%) was higher than that of SP and SBE oils (average values 17.70% and 17.92%, respectively), while the C18:2n-6 content (average value 62.00%) was lower than that of the other two (average values 63.70% and 63.03%, respectively). The result accorded with the rule that C18:1n-9 and C18:2n-6 transformed each other in the process of walnut oil ripening, and their sum was equal (Gao et al., 2021). The C18:2n-6 was the main component of PUFA, MR-SP had the highest C18:2n-6 content (64.95%), and, although the C18:3n-3 content of MR-SP (8.75%) was significantly lower than that of other samples, its PUFA content (73.69%) was the highest among all walnut oils.

### 3.4. Phytochemicals Content

Table 2 lists the phytochemicals present in walnut oil produced by different pretreatment and processing methods, which included three forms of tocopherol (α, γ and δ), phytosterols, squalene and TPC. The α, γ, δ and total tocopherol contents of all samples were significantly higher than those of the control, which proved that pretreatment and processing could improve the content of tocopherol in walnut oil. For walnut oil with the same processing method, the content of α-tocopherol was MR (17.76–18.84 mg/kg) > OR (16.94–18.62 mg/kg) > SR (13.73–17.56 mg/kg) in descending order according to the pretreatment method, and the content of SR was significantly lower than that of the other two groups. γ-Tocopherol was the major tocopherol in walnut oil, and MR-SP had the highest content (359.90 mg/kg) of all walnut oils, which was 96.42 mg/kg higher than OR-SP. Therefore, its total tocopherol content (419.85 mg/kg) was also the highest, which was 27.41% higher than that of OR-SP. There were significant differences in the δ-tocopherol contents of walnut oils in different processing methods, and the contents were: SBE (45.88–46.71 mg/kg) > AEE (42.97–44.70 mg/kg) > SP (38.36–42.19 mg/kg). Tocopherol is very sensitive and might decompose in an environment subjected to heat and solvents [20].

Table 2 also presents the phytosterols, squalene and TPC contents of walnut oil samples. The pretreatment method would reduce the phytosterols content of SP oils, which was significantly lower than that of the control (1474.18 mg/kg). The SR-SBE oil had the highest content, 248.69 mg/kg higher than that of the lowest OR-SP oil. This might be because phytosterols undergo oxidation, isomerization, dehydroxylation, hydrolysis, dehydrogenation and other intermolecular transformation reactions during processing, resulting in significant differences in their contents. In addition, the content of squalene in walnut oil was low (7.65–9.91 mg/kg), while pretreatment methods could significantly increase the content in SP samples. The average squalene content of AEE oils (8.00 mg/kg) was lower than that of the other two; the average contents of SP and SBE were both 9.46 mg/kg. In addition, the results of TPC were similar to those of squalene, pretreatment technology had little influence on the TPC content of walnut oil, and the content had a decreasing order of SP > SBE > AEE > control. MR-SP oil had the highest content (13.12 mg/kg), which was 6.64 mg/kg higher than that of the control.

### 3.5. Antioxidant Activity

The OSI could be used to predict the shelf life of oil, and the free radical scavenging capacity could evaluate the antioxidant activity of oil from a physiological point of view. The OSIs and free radical scavenging capacities of walnut oils are shown in Table 3. The OSIs of different walnut oil samples were significantly higher than that of the control (1.80 h), and MR-SP oil had the best oxidative stability (2.63 h). According to the detection rules of the oxidation stabilizer, the oxidation rate was doubled every 10 degrees. Therefore, the results here are better than those of our previous paper [10].

The DPPH free radical scavenging capacity of the samples was assessed by whole oil, nonpolar and polar extract parts. Except for that of OR-AEE oil (128.07 μmol TE/kg), the DPPH-oil free radical scavenging capacity of all walnut oils was better than that of the control. The DPPH-oil free radical scavenging capacities with different processing methods were significantly different, and AEE oils featured lower capacities than that of others. In contrast, pretreatment methods would reduce the DPPH-nonpolar extract free radical scavenging capacity of SP oils. The free radical scavenging capacities of SP samples (72.59–76.83 μmol TE/kg) were significantly lower than that of the control (91.64 μmol TE/kg). The pretreatment method of DPPH-polar free radical scavenging capacity in SP oils also showed a negative influence, which increased in the order of SP (30.69–44.70 μmol TE/kg) < control (105.40 μmol TE/kg) < AEE (163.04–248.76 μmol TE/kg) < SBE (287.59–373.14 μmol TE/kg). MR-SBE oil with the strongest DPPH-polar free radical scavenging capacity was 12 times higher than that of MR-SP oil, which had the worst effect.

The trend of walnut oil ABTS free radical scavenging capacity was consistent with the DPPH-polar results, which decreased in the order of SBE (269.34–280.28 μmol TE/kg) > AEE (239.51–260.39 μmol TE/kg) > control (233.54 μmol TE/kg) > SP (206.70–231.06 μmol TE/kg). This might be because both DPPH-polar and ABTS assess the ability to quench free radicals by hydrogen donation, which is according to hydrogen atom transfer methods [20]. Generally, the scavenging reaction between radicals and antioxidants present in the oil starts with the transfer of the most labile H atom from the scavenger molecule to the free radical [36]. The FRAP free radical scavenging capacity of AEE oil was lower than that of others, and the OR-AEE oil was the lowest (50.25 μmol TE/kg). MR-SP oil had the highest capacity (105.61 μmol TE/kg) of FRAP, which was 23.52% higher than that of the control and 55.36 μmol TE/kg higher than that of OR-AEE oil.

### 3.6. HCA Analysis

HCA was used to evaluate the similarities among walnut oils. These results are presented as a dendrogram in Figure 1, which is utilized to convey a hierarchy based on the similarities among different pretreatment and process samples. When the 25-distance threshold was selected, the tree structure of the cluster analysis was divided into two main parts, and walnut oil from SP and other process methods could be clearly distinguished. The results revealed that the processing method significantly affected the chemical components and antioxidant capacity of walnut oil. AEE technology used biological enzyme preparation to destroy or dissolve the cell wall and oil complex after mechanical crushing, and then the oil was extracted by the characteristics of the immiscibility of oil and water [37], while SBE technology used butane as an extraction solvent, according to the principle of similarity and intervisibility [38]. Although the principles of AEE and SBE were different, both of them belonged to extraction technology, which was much different with the SP method. Therefore, we could conclude that processing method had the greatest influence on the characterizations and antioxidant capacity of walnut oil. In addition, the walnut oils obtained with MR and OR pretreatment were more similar in composition and character. This might be because although the heat conduction rates of MR and OR are different, their principles are similar. In heating or roasting, thermal energy reached the surface of walnut kernels by radiation or convection heating, which was then transferred gradually to the bulk of the kernel via conduction [39]. Thus, the contents of phytochemicals and their antioxidant properties were affected in walnut oil.

### 3.7. MLR Analysis

As some chemical components of walnut oil might exhibit antioxidant activity, MLR was used to shed light on a certain correlation and mutual restriction between overall antioxidant capacity and that of single or multiple chemical components (Table 4). The OSI and DPPH-oil models showed 0.727 and 0.711 of adjusted R^2^, with the partial correlation coefficients of the predicted equation equaling 0.870 and 0.862 for TPC, respectively. The obtained results indicated that the antioxidant activity of whole oil was related only to TPC. OSI and DPPH-oil were two different types of indicators expressing the antioxidant capacity of walnut oil. Our previous research (Gao et al., 2018; Gao et al., 2021) reported that the antioxidant capacity of walnut oil was related to TPC, which was demonstrated again.

The nonpolar extracts in walnut oil were mainly triglycerides and free fatty acids; thus, C18:0 and C18:3n-3 were the significantly independent variables of DPPH-nonpolar assay. In addition, the model constructed using the results of the ABTS assay showed the highest regression coefficient (R^2^ = 0.954) and C18:0, C18:1n-9, δ-tocopherol and γ-tocopherol contents were correlated as Y = 0.647 (C18:0) − 0.657 (C18:1n-9) + 0.942 (δ-tocopherol) − 0.601 (γ-tocopherol). Furthermore, γ-tocopherol exhibited a significant negative contribution to DPPH-polar, while δ-tocopherol (R = 1.260) played an important positive role. Moreover, C18:3n-3 (R = 0.395) influenced the FRAP radical scavenging activity of the samples.

C18:3n-3 was the highest unsaturated fatty acid in walnut oil, and the double bond contained in C18:3n-3 was the main reason for lipid oxidation because the double bond was easy to open and became a free radical receptor under light, heat or other oxidation conditions, which intensified the oxidation reaction; thus, C18:0 could inhibit oxidation. The phenolic hydroxyl groups in tocopherol structure were the main sources of its antioxidant activity, but the methyl substitution of hydroxyl groups at different positions would affect the antioxidant activity and steric hindrance of tocopherol monomers. Therefore, γ-tocopherol and δ-tocopherol showed opposite antioxidant effects in walnut oil.

## 4. Conclusions

This study investigated the effects of different pretreatment and processing on the chemical properties and quality characteristics of walnut oil. The oil yield of SBE oils (59.53–61.19%) was higher than that of other processing technologies, which also had better DPPH-polar (287.59–373.14 μmol TE/kg) and ABTS (269.34–280.28 μmol TE/kg) free radical scavenging ability. Owing to higher oil yield and better antioxidant activity, SBE processing is recommended for extracting oil from walnuts for various applications in industry. The SR-AEE process resulted in higher acid value (2.49 mg/g) and peroxide value (4.16 mmol/kg), while the MR-SP process would increase the content of PUFA (73.69%), total tocopherol (419.85 mg/kg) and total phenolic compounds (TPC, 13.12 mg/kg). HCA results showed that processing technology formed the main factor affecting the chemical characterizations of walnut oil. Moreover, MLR confirmed that C18:0, C18:1n-9, C18:3n-3, γ- tocopherol, δ-tocopherol and TPC were strongly correlated with the antioxidant capacity of walnut oil.

## Figures and Tables

**Figure 1 foods-11-01698-f001:**
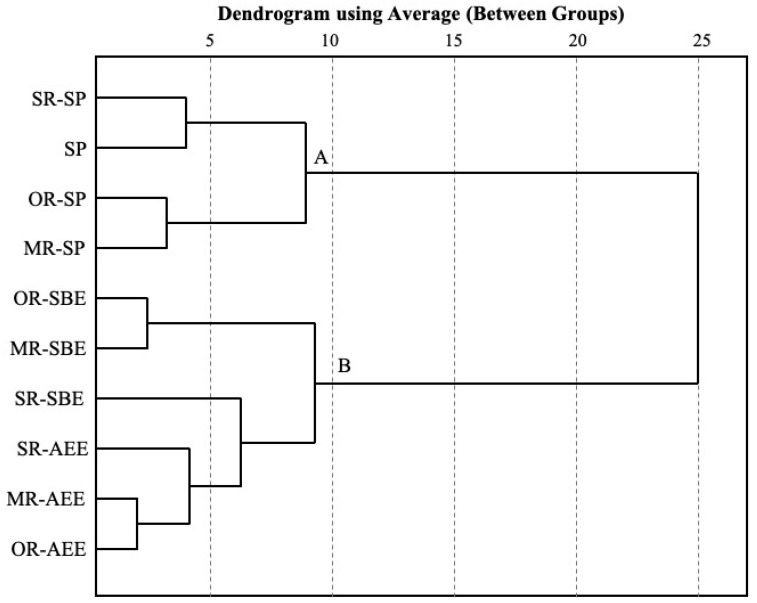
Hierarchical cluster analysis of walnut oils.

**Table 1 foods-11-01698-t001:** The physiochemical indexes and fatty acid compositions (%) of walnut oils.

	SP	MR-SP	OR-SP	SR-SP	MR-AEE	OR-AEE	SR-AEE	MR-SBE	OR-SBE	SR-SBE
Lipid yield (%)	46.31±0.55 ^f^ *	40.73 ± 0.41 ^h^	40.79 ± 0.17 ^h^	43.29 ± 0.71 ^g^	53.65 ± 0.33 ^c^	52.73 ± 0.31 ^d^	50.64 ± 0.39 ^e^	60.75 ± 0.30 ^a^	61.19 ± 0.25 ^a^	59.53 ± 0.44 ^b^
Acid value (mg/g)	0.45 ± 0.10 ^cd^	0.57 ± 0.00 ^c^	0.14 ± 0.02 ^f^	0.33 ± 0.01 ^de^	1.12 ± 0.03 ^b^	1.14 ± 0.01 ^b^	2.49 ± 0.16 ^a^	0.26 ± 0.01 ^ef^	0.23 ± 0.00 ^ef^	0.23 ± 0.00 ^ef^
Peroxide value (mmol/kg)	5.56 ± 0.07 ^a^	3.38 ± 0.34 ^c^	2.63 ± 0.24 ^de^	2.45 ± 0.02 ^e^	3.37 ± 0.08 ^c^	2.52 ± 0.18 ^de^	4.16 ± 0.11 ^b^	2.33 ± 0.02 ^e^	2.61 ± 0.05 ^de^	2.81 ± 0.04 ^d^
Fatty acid (%)										
C16:0	6.41 ± 0.02 ^a^	6.14 ± 0.00 ^f^	6.18 ± 0.02 ^e^	6.19 ± 0.00 ^de^	6.21 ± 0.00 ^d^	6.21 ± 0.01 ^d^	6.25 ± 0.00 ^c^	6.23 ± 0.02 ^c^	6.21 ± 0.01 ^d^	6.30 ± 0.01 ^b^
C18:0	2.75 ± 0.00 ^abc^	2.62 ± 0.01 ^e^	2.70 ± 0.01 ^d^	2.70 ± 0.02 ^d^	2.77 ± 0.02 ^a^	2.76 ± 0.01 ^ab^	2.74 ± 0.00 ^bc^	2.76 ± 0.01 ^abc^	2.74 ± 0.01 ^c^	2.75 ± 0.01 ^abc^
C18:1n-9	18.51 ± 0.03 ^b^	17.36 ± 0.04 ^i^	17.70 ± 0.02 ^g^	18.03 ± 0.02 ^f^	18.84 ± 0.02 ^a^	18.53 ± 0.02 ^b^	18.35 ± 0.02 ^c^	18.10 ± 0.02 ^e^	18.15 ± 0.01 ^d^	17.51 ± 0.02 ^h^
C18:2n-6	62.10 ± 0.01 ^h^	64.95 ± 0.06 ^a^	63.21 ± 0.02 ^c^	62.94 ± 0.01 ^d^	61.73 ± 0.02 ^j^	61.98 ± 0.03 ^i^	62.31 ± 0.02 ^g^	62.55 ± 0.02 ^f^	62.64 ± 0.01 ^e^	63.89 ± 0.02 ^b^
C18:3n-3	10.04 ± 0.02 ^de^	8.75 ± 0.01 ^h^	10.01 ± 0.01 ^e^	9.95 ± 0.01 ^f^	10.26 ± 0.02 ^b^	10.33 ± 0.05 ^a^	10.17 ± 0.03 ^c^	10.17 ± 0.01 ^c^	10.08 ± 0.02 ^d^	9.36 ± 0.02 ^g^
C20:1	0.18 ± 0.00 ^a^	0.19 ± 0.00 ^a^	0.19 ± 0.01 ^a^	0.19 ± 0.00 ^a^	0.19 ± 0.01 ^a^	0.19 ± 0.01 ^a^	0.19 ± 0.00 ^a^	0.19 ± 0.01 ^a^	0.18 ± 0.01 ^a^	0.19 ± 0.00 ^a^
SFA ^#^	9.16 ± 0.02 ^a^	8.76 ± 0.01 ^f^	8.89 ± 0.01^e^	8.89 ± 0.02 ^e^	8.98 ± 0.02 ^c^	8.97 ± 0.02 ^cd^	8.99 ± 0.00 ^c^	8.99 ± 0.02 ^c^	8.94 ± 0.01 ^d^	9.05 ± 0.01 ^b^
MUFA	18.69 ± 0.03 ^b^	17.55 ± 0.04 ^i^	17.89 ± 0.01 ^g^	18.22 ± 0.02 ^f^	19.03 ± 0.02 ^a^	18.71 ± 0.02 ^b^	18.54 ± 0.02 ^c^	18.29 ± 0.03 ^e^	18.34 ± 0.02 ^de^	17.70 ± 0.02 ^h^
PUFA	72.15 ± 0.01 ^g^	73.69 ± 0.05 ^a^	73.22 ± 0.02 ^b^	72.89 ± 0.00 ^c^	71.99 ± 0.03 ^h^	72.32 ± 0.03 ^f^	72.47 ± 0.02 ^e^	72.72 ± 0.03 ^d^	72.72 ± 0.02 ^d^	73.25 ± 0.01 ^b^

* Values are means ± standard deviations. The superscript letters indicate the statistical differences in rows at significance level of 5%. ^#^ SFA: C14:0 + C16:0 + C18:0, MUFA: C16:1 + C18:1 + C20:1, PUFA: C18:2 + C18:3.

**Table 2 foods-11-01698-t002:** The phytochemicals content (mg/kg) of walnut oils.

	α-Tocopherol	γ-Tocopherol	δ-Tocopherol	Total Tocopherol	Phytosterols	Squalene	TPC
SP	5.64 ± 0.22 ^g^ *	244.67 ± 14.49 ^f^	35.64 ± 0.21 ^g^	285.95 ± 14.40 ^e^	1474.18 ± 6.31 ^d^	8.39 ± 0.01 ^g^	6.48 ± 0.02 ^e^
MR-SP	17.76 ± 0.11 ^b^	359.90 ± 15.02 ^a^	42.19 ± 0.09 ^e^	419.85 ± 15.01 ^a^	1462.65 ± 3.34 ^e^	9.60 ± 0.01 ^c^	13.12 ± 0.05 ^a^
OR-SP	16.94 ± 0.35 ^f^	263.48 ± 7.12 ^e^	38.36 ± 0.43 ^h^	304.78 ± 7.14 ^d^	1361.36 ± 4.62 ^g^	9.91 ± 0.01 ^a^	12.27 ± 0.12 ^ab^
SR-SP	13.73 ± 0.17 ^e^	313.36 ± 3.00 ^cd^	39.64 ± 0.27 ^f^	366.73 ± 3.10 ^bc^	1367.10 ± 1.31 ^g^	8.86 ± 0.01 ^f^	10.59 ± 0.14 ^c^
MR-AEE	17.76 ± 0.02 ^b^	316.42 ± 3.50 ^c^	44.61 ± 0.43 ^c^	378.79 ± 3.42 ^b^	1394.32 ± 3.30 ^g^	8.12 ± 0.02 ^i^	9.64 ± 0.10 ^d^
OR-AEE	17.41 ± 0.09 ^c^	296.60 ± 3.31 ^d^	44.70 ± 0.09 ^c^	358.71 ± 3.29 ^c^	1416.75 ± 1.91 ^f^	8.24 ± 0.03 ^h^	8.13 ± 0.07 ^c^
SR-AEE	15.60 ± 0.13 ^d^	318.93 ± 6.40 ^cd^	42.97 ± 0.60 ^d^	377.50 ± 6.31 ^b^	1557.00 ± 9.21 ^b^	7.65 ± 0.02 ^j^	10.45 ± 0.16 ^ab^
MR-SBE	18.84 ± 0.04 ^a^	305.25 ± 3.19 ^cd^	45.88 ± 0.30 ^b^	369.97 ± 2.99 ^bc^	1560.01 ± 8.19 ^b^	9.77 ± 0.03 ^b^	10.30 ± 0.11 ^ab^
OR-SBE	18.62 ± 0.10 ^a^	305.68 ± 3.86 ^c^	46.71 ± 0.59 ^a^	371.00 ± 4.24 ^bc^	1489.75 ± 4.32 ^c^	9.10 ± 0.02 ^e^	12.16 ± 0.10 ^ab^
SR-SBE	17.56 ± 0.17 ^bc^	340.23 ± 3.81 ^b^	46.17 ± 0.44 ^b^	403.96 ± 4.09 ^a^	1610.05 ± 7.69 ^a^	9.50 ± 0.02 ^d^	11.22 ± 0.05 ^b^

* Values are means ± standard deviations. The superscript letters indicate the statistical differences in lines at significance level of 5%.

**Table 3 foods-11-01698-t003:** Oxidation stability indexes (h) and free radical scavenging capacities (μmol TE /kg) of walnut oils.

	OSI	DPPH-Oil	DPPH-Nonpolar	DPPH-Polar	ABTS	FRAP
SP	1.80 ± 0.05 ^e^ *	132.93 ± 4.63 ^de^	91.64 ± 3.97 ^cd^	105.40 ± 10.56 ^e^	233.54 ± 3.72 ^def^	85.50 ± 5.01 ^bc^
MR-SP	2.63 ± 0.08 ^a^	171.70 ± 12.20 ^abc^	73.26 ± 1.30 ^f^	30.69 ± 2.48 ^f^	206.70 ± 3.06 ^g^	105.61 ± 5.13 ^a^
OR-SP	2.53 ± 0.06 ^ab^	193.19 ± 4.05 ^a^	76.83 ± 3.02 ^ef^	43.40 ± 4.92 ^f^	227.08 ± 1.22^f^	67.22 ± 5.95 ^e^
SR-SP	2.34 ± 0.05 ^c^	157.76 ± 17.96 ^bc^	72.59 ± 7.79 ^f^	44.70 ± 6.31 ^f^	231.06 ± 5.76 ^ef^	84.79 ± 3.54 ^bc^
MR-AEE	2.14 ± 0.06 ^d^	153.92 ± 7.77 ^cd^	108.36 ± 3.59 ^a^	248.76 ± 9.54 ^c^	241.50 ± 3.06 ^d^	50.49 ± 5.60 ^f^
OR-AEE	2.32 ± 0.02 ^c^	128.07 ± 9.33 ^e^	101.96 ± 5.86 ^abc^	245.03 ± 24.71 ^c^	260.39 ± 0.70 ^c^	50.25 ± 4.88 ^f^
SR-AEE	2.51 ± 0.03 ^ab^	147.11 ± 13.64 ^de^	94.88 ± 9.22 ^bcd^	163.04 ± 18.90 ^d^	239.51 ± 9.30 ^de^	69.49 ± 7.05 ^de^
MR-SBE	2.52 ± 0.04 ^ab^	158.22 ± 11.38 ^bc^	107.01 ± 4.23 ^ab^	373.14 ± 20.78 ^a^	269.34 ± 4.28 ^b^	80.97 ± 3.94 ^bcd^
OR-SBE	2.51 ± 0.08 ^ab^	180.54 ± 7.76 ^ab^	87.15 ± 10.40 ^de^	341.24 ± 21.02 ^a^	275.80 ± 2.54 ^ab^	91.81 ± 7.28 ^b^
SR-SBE	2.48 ± 0.03 ^b^	181.67 ± 7.78 ^ab^	109.46 ± 2.85 ^a^	287.59 ± 20.48 ^b^	280.28 ± 1.86 ^a^	77.85 ± 1.54 ^cde^

* Values are means ± standard deviations. The superscript letters indicate the statistical differences in lines at significance level of 5%.

**Table 4 foods-11-01698-t004:** Equations, variables and regression coefficients in the prediction of antioxidant capacity by multiple linear regression.

Dependent Variable	Adjusted R^2^	Variable	Standardized Coefficient	Significance (Two Tails *p*)	Equation
OSI	0.727	TPC	0.870	0.001	Y = 0.870 (TPC)
DPPH-oil	0.711	TPC	0.862	0.001	Y = 0.862 (TPC)
DPPH-nonpolar	0.772	C18:0	1.344	0.001	Y = 1.344 (C18:0) − 0.689 (C18:3)
		C18:3	−0.689	0.032	
DPPH-polar	0.910	δ-tocopherol	1.260	0.000	Y = 1.260 (δ-tocopherol) − 0.699 (γ-tocopherol)
		γ-tocopherol	−0.699	0.001	
ABTS	0.954	C18:0	0.647	0.007	Y = 0.647 (C18:0) − 0.657 (C18:1) + 0.942 (δ-tocopherol) − 0.601 (γ-tocopherol)
		C18:1	−0.657	0.002	
		δ-tocopherol	0.942	0.002	
		γ-tocopherol	−0.601	0.016	
FRAP	0.395	C18:3	−0.680	0.030	Y = −0.680 (C18:3)

## Data Availability

No new data were created or analyzed in this study. Data sharing is not applicable to this article.

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
