# Peer review of "Characteristics and Antioxidant Activity of Walnut Oil Using Various Pretreatment and Processing Technologies"

_foods, 2022, doi:10.3390/foods11121698_

Round 1

Reviewer 1 Report

It is not clear why the working conditions tested are used.

It is necessary to support the results obtained with bibliography.

The rest of the comments can be found in the attached PDF.

Author Response

Dear Editors and Reviewers:

Thank you very much for the useful suggestions for our manuscript entitled Characteristics and antioxidant activity of walnut oil using various pretreatment and processing technologies” (ID: foods-1760421) Those comments are all valuable and very helpful for revising and improving our paper, as well as the important guiding significance to us researches. We have studied the comments carefully and revised the manuscript which we hope meet with approval. The detailed responses to the reviews’ comments are in a single file. The revised portion are marked in red in the revised manuscript.

We hope that the revised manuscript is now acceptable for publication. We look forward to hearing from you. If you have any question, please do not hesitate to contact me.

Thank you.

Best regards and happy new year!

Sincerely yours,

Pan Gao

[email protected]

Reviewer 2 Report

foods-1760421-peer-review-v1.comments

Characteristics and antioxidant activity of walnut oil using various pretreatment and processing 2 technologies

Pan Gao1,2*, Yunpeng Ding1, Zhe Chen2, Zhangtao Zhou1, Wu Zhong1,2, Chuanrong Hu1, Dongping 4 He1,2, Xingguo Wang1,3 5

Comment 1: Line No. The study compared the results of pretreatment, processing technology and quality of walnut oil seems to be very informative.  

Comment 2: Line No. 52 & 53: Do this prevention of memory impairment in mice was linked with anti-inflammatory behavior only?

Comment 3: Line No. 57 & 58: Now here does this walnut oil has showed its functional behavior towards inflammatory bowel disease and ulcerative colitis; its anti-aging and anti-oxidative properties only. Rather I would suggest a link here that the anti-inflammatory action is more dominant in case of ameliorating the inflammatory bowel disease and ulcerative colitis. And for above mentioned health outcome of preventing memory impairment, the anti-aging and anti-oxidative potentials would serve more beneficial roles.

Comment 4: Line No. 46 & 52 and so on: Avoid unnecessary line spacing/gapping throughout the manuscriot 

Comment 5: Line No. 52: In my opinion after first paragraph on background for the study, there must be a couple of paragraphs on nutritional composition, phytochemical profile, and stability aspects of walnut oil.

Comment 6: Line No. 73: Can you please elaborate what kind of toxins and anti-nutritional compounds have been addressed here?

Comment 7: Line No. 106:  Clearly identify the objectives of study at the last paragraph here, and also justify the reason for selecting particular treatments and techniques only i.e. microwave roasting, oven roasting, steaming roasting, screw-pressing, aqueous enzymatic extraction, subcritical butane extraction..

Comment 8: Line No. Rest the whole manuscript is overall well written, results and discussion section has been thoroughly covering required stuff, and therefore the manuscript is recommended slight amendments at the introductory section.

Author Response

(The authors gave the same response as above.)

Round 2

Reviewer 1 Report

The answers to the comments made are correct

Reviewer 2 Report

Well done.